# Genomic Tools for the Characterization of Local Animal Genetic Resources: Application in *Mascaruna* Goat

**DOI:** 10.3390/ani12202840

**Published:** 2022-10-19

**Authors:** Marco Tolone, Maria Teresa Sardina, Gabriele Senczuk, Giorgio Chessari, Andrea Criscione, Angelo Moscarelli, Silvia Riggio, Ilaria Rizzuto, Rosalia Di Gerlando, Baldassare Portolano, Salvatore Mastrangelo

**Affiliations:** 1Dipartimento Scienze Agrarie, Alimentari e Forestali, University of Palermo, 90128 Palermo, Italy; 2Dipartimento di Agricoltura, Ambiente e Alimenti, University of Molise, 86100 Campobasso, Italy; 3Dipartimento Agricoltura, Alimentazione e Ambiente, University of Catania, 95131 Catania, Italy

**Keywords:** local goat, single nucleotide polymorphisms, genetic structure

## Abstract

**Simple Summary:**

The Analysis of genomic data is an important resource for the effective management of small and endangered populations. The aim of this work was to study the genomic structure of a local goat named Mascaruna. Overall, the results indicate moderate genetic variability and a defined population structure and represent a starting point for the creation of monitoring and conservation plans.

**Abstract:**

Italy contains a large number of local goat populations, some of which do not have a recognized genetic structure. The “Mascaruna” is a goat population reared for milk production in Sicily. In this study, a total of 72 individuals were genotyped with the Illumina Goat_IGGC_65K_v2 BeadChip with the aim to characterize the genetic diversity, population structure and relatedness with another 31 Italian goat populations. The results displayed a moderate level of genetic variability for Mascaruna, in concordance with the estimated values for Italian goats. Runs of homozygosity islands are linked to genes involved in milk production, immune response and local adaptation. Population structure analyses separated Mascaruna from the other goat populations, indicating a clear genetic differentiation. Although they are not conclusive, our current results represent a starting point for the creation of monitoring and conservation plans. Additional analyses and a wider sampling would contribute to refine and validate these results. Finally, our study describing the diversity and structure of Mascaruna confirms the usefulness of applied genomic analyses as valid tools for the study of the local uncharacterized genetic resources.

## 1. Introduction

Goats (*Capra hircus*) are economically important domestic animals reared for milk, meat and fiber production. Italy is the European country with the highest number of local goat populations and, therefore, a precious reservoir of genetic diversity [1,2]. Indeed, the environmental diversity and socio-economic factors led to the formation of a large number of local populations, some of which were then standardized into modern breeds due to isolation and selection processes. On the other hand, several goat populations are still locally reared, especially in the southern regions [3,4]. The majority of these populations are kept by smallholders under extensive production systems, are well adapted to the local environments and represent an important economic source in marginal areas. Their genetic diversity and conservation is an important aspect in the light of facing future challenges (e.g., climate change and emerging diseases) [5]. Therefore, the genetic characterization is desirable for these populations not only to trace their origin but also to assess their inbreeding and to provide information in establishing proper management and conservation actions [6].

Several local goats are reared in Sicily, some of which do not have a defined genetic structure, being the result of crosses between populations sharing the same geographic area [7]. An interesting situation is represented by “Mascaruna”, a goat population reared for milk production. The origin of this population is unknown. A previous study on the genetic characterization, through microsatellite markers and a low number of individuals, outlined the hypothesis that Mascaruna is not an admixed goat but a population with a defined genetic background [8]. The animals are phenotypically homogeneous: they present with a medium-high size, cream color with a dark brown line on the back, and long hair type coat (Figure 1). The abdomen and the legs are also dark brown. In both sexes, the head is light and presents with small ears and dark brown elongated spots around the eyes.

By definition, breed or population means a group of closely related individuals possessing uniform traits (e.g., coat color), restricted gene flow and higher inbreeding values compared to what is expected from outbred individuals [9]. Therefore, the genetic characterization of this goat is needed to start a process of official recognition as a breed.

The findings from genomic data have become an invaluable resource for effective management of breeding programs in small and endangered populations [10]. The availability of a medium density single nucleotide polymorphism (SNP) array for goat species [11] has provided the opportunity to improve the knowledge on the genetic differentiation between goat populations, in particular for local genetic resources [12,13,14]. The Italian caprine diversity was previously studied through SNP data analyses [1,2]. However, some local goats such as Mascaruna remain uncharacterized. In fact, the knowledge of the population structure of this goat and its divergence from the other Italian populations is still lacking.

In this study, we used genomic tools to genotype the Mascaruna goat with the aim to characterize the genetic diversity, population structure and relationships with other Italian goat populations and to widen the picture of the genome-wide diversity of Italian caprine species.

## 2. Materials and Methods

### 2.1. DNA Sampling

A total of 72 Mascaruna goat DNA samples (62 females and 10 males) were genotyped. DNA was extracted from blood using the commercial Illustra blood genomic Prep Mini Spin kit (GE Healthcare, Little Chalfont, UK). Animals were chosen on the basis of their phenotypic profile (morphological traits such as coat color) and the information provided by farmers in order to collect unrelated individuals.

### 2.2. Genotyping, Quality Control and Data Handling

All animals were genotyped with the Illumina Goat_IGGC_65K_v2 BeadChip containing 59,727 SNPs (Illumina, San Diego, CA, USA). The raw data of Mascaruna goat were merged with the genotypic data of Italian Goat Consortium2 (IGC2) (Table 1) as described in Cortellari et al. [2], obtaining a final dataset consisting of 975 individuals, 50,641 common SNPs (between the v1 and v2 Goat BeadChip) and 32 populations (hereafter called 32POP dataset). Moreover, to investigate in detail the relationship between the Mascaruna and the other goat populations from southern Italy, a reduced dataset was also created (hereafter called 8POP dataset) (Table 1).

The genomic coordinates for each marker were obtained using the assembled goat genome (ARS1, GCA_001704415.1). SNPs without position or assigned to the X chromosome were discarded. The software PLINK v. 1.9 [15] was used to perform filtering and quality control using the following criteria: a minor allele frequency ≥0.05, a genotype call rate for a SNP ≥0.95 and an individual call rate ≥0.90. After quality check, 963 animals and 48,586 SNPs (32POP) and 306 animals and 48,347 SNPs (8POP) were retained, respectively.

### 2.3. Genetic Diversity Indices and Inbreeding

After applying the above quality filters, all the analyses within the Mascaruna goat population were conducted using 67 individuals and 50,291 SNPs. PLINK v. 1.9 [15] was used to estimate observed (H_O_) and expected (H_E_) heterozygosities, inbreeding coefficient (F_IS_) and the minor allele frequencies (MAF) in order to compare the results with those reported in the literature for all Italian goat populations [16]. Moreover, the contemporary effective population size (CNe) was estimated with NEESTIMATOR v. 2 [17] by using the random mating model of the linkage disequilibrium (LD) method. In addition, we performed the runs of homozygosity (ROH) analysis to obtain an estimate of the molecular inbreeding by using PLINK v. 1.9 [15] and following the parameters reported by Cortellari et al. [16]. The inbreeding coefficient (F) based on ROH (F_ROH_) for each animal, the mean number of ROH per individual, the mean total length of ROH and the mean length of ROH per individual were estimated.

### 2.4. Runs of Homozygosity Islands in Mascaruna Goat

Highly homozygous genomic regions (ROH islands) were also identified. The percentage of SNPs present in ROH was calculated based on their frequency across individuals [18]. The top 0.999 SNPs of the percentile distribution were selected to form ROH islands. Annotated genes within the ROH islands were examined using the Genome Data Viewer tool provided by NCBI (https://www.ncbi.nlm.nih.gov/genome/gdv?org=capra-hircus&group=bovidae accessed on 4 July 2022) and a precise literature research.

### 2.5. Genetic Relationship and Population Structure between Mascaruna and Other Goat Populations

In order to understand the genetic relationships and the population structure, the datasets were also filtered to remove SNPs in high LD (r2 > 0.2) by using the --indep-pairwise (50 10 0.2) function in PLINK v. 1.9 [15], generating a pruned dataset of 43,625 SNPs for 32POP and 42,087 SNPs for 8POP, respectively.

The genetic relationships among populations were estimated using an identity-by-state (IBS) matrix of genetic distances calculated by PLINK v. 1.9 [15] and plotted using a multidimensional scaling (MDS) plot in the R environment. ARLEQUIN v. 3.5 software [19] was used to estimate Reynolds genetic distances, and the neighbor-net tree was constructed using SPLITSTREE v. 4.14.8 [20]. ARLEQUIN v 3.5 was also used to estimate population relatedness using pairwise estimates of F_ST_. All these analyses were carried out both on 32POP and 8POP datasets. Patterns of ancestry and admixture were examined by the model-based clustering algorithm implemented in the ADMIXTURE software v1.3.0 [21], by applying the default settings and different K values (K = 2 to 30) in the 32POP dataset. The most likely number of clusters was estimated following the cross-validation procedure. The results were plotted using the *membercoef.circos* function in the R package BITE [22]. In order to explore migration events, a maximum likelihood dendogram was generated using the TREEMIX software v. 1.13 [23]. For this analysis, we used the genotypic data of Bezoar as outgroup. Five independent iterations were performed allowing migration events to range between 1 and 10, while the covariance matrix was estimated using 500 contiguous SNPs per block. The most supported number of migration edges was assessed using the linear method as implemented in the R package OptM [24].

## 3. Results

### 3.1. Genetic Diversity Indices

Genetic diversity indices, estimated using different approaches, were adopted to identify the levels of genetic variability in Mascaruna goat. Descriptive statistics are reported in Table 2 and displayed a moderate level of genetic variability.

In total, 2381 ROHs were identified, ranging from 4 to 101 ROH per individual. The average number of ROH per animal was 35.5, with an average length of 4.85 Mb. The majority of ROH segments (79.25%) were shorter than 8 Mb in length, while 166 segments (~7%) were longer than 16 Mb. The relationship between total number of ROH and the total genomic length for each animal showed that most of the individuals (64%) led from 4 to 40 ROH with a total length <200 Mb. There were animals with a large number of ROH (above 70), with more than 500 Mb of total length of genome in ROH (Appendix A).

### 3.2. Runs of Homozygosity Islands in Mascaruna Goat

The top 0.999 SNPs of the percentile distribution were selected, and adjacent SNPs over this threshold were merged into genomic regions corresponding to ROH islands in Mascaruna (Figure 2). In the Manhattan plot, each significant marker showed a percentage of occurrence >25%, and few peaks above this minimum value were identified.

Table 3 provides the chromosome position, start and end and the number of SNPs of these regions with the annotated genes. In total, we detected four ROH islands, one on the *Capra hircus* chromosome (CHI) 2 and CHI6, respectively, and two on CHI12. These regions ranged from 580 kb (CHI2) to 1.83 Mb (CHI6). We identified a total of 18 known genes together with uncharacterized loci (LOC). The region on CHI6 contained, among the others, 13 SNPs with the highest H value (45%) (Figure 2).

### 3.3. Genetic Relationship and Population Structure between Mascaruna and Other Goat Populations

To examine the genetic relationships within and between populations, we used an MDS plot of the pairwise IBS distances (32POP). The first and second dimensions explained 4.29% and 2.65% of the observed variation, respectively (Figure 3). The results showed that most populations formed non-overlapping clusters, and they were clearly separated. In particular, the first dimension (C1) distinguished the southern from the northern goat populations, with the central Italian goats positioned between the two clusters. The second dimension (C2) separated the Girgentana from the other Italian goats. The Mascaruna individuals clustered together with the Sicilian and southern goat populations (Appendix A).

The MDS plot was also created only for the southern goat populations (8POP) (Figure 4). The results indicated a moderate genetic isolation for Mascaruna with a defined cluster. The Sicilian goats (Argentata dell’Etna, Messinese, Maltese and Rossa Mediterranea) confirmed their proximity. In particular, the first dimension (7.78%) separated the Girgentana breed from the other populations, whereas the second dimension (5.34%) discriminated the Mascaruna from the southern populations. In this reduced context, the Mascaruna also showed a more widespread cluster (Figure 4).

The pairwise F_ST_ value among all 32 populations ranged from 0.001 (Messinese vs. Argentata dell’Etna) to 0.163 (Orobica vs. Maltese). Moderate genetic differentiation was observed between Mascaruna and several Italian goat populations. Genetic differentiation ranged from 0.037 (Mascaruna vs. Argentata dell’Etna) to 0.12 (Mascaruna vs. Orobica and Vallesana breeds) (Appendix A, 32POP). Considering only the relationship between Mascaruna and southern Italian goats, pairwise F_ST_ values ranged from 0.036 (Mascaruna vs. Argentata dell’Etna) to 0.086 (Mascaruna vs. Girgentana) (Appendix A, 8POP).

To provide additional indications regarding the relationships and the origin of Mascaruna, we depicted a neighbor-net graph based on Reynolds genetic distances (Figure 5, 32POP).

The graph showed a clear clusterization among populations that originated from the same geographic area, confirming a north-to-south genetic pattern from right to left. Mascaruna branched with Girgentana, Aspromontana, Messinese and Argentata dell’Etna, grouped in one cluster in which Girgentana and Aspromontana showed a closer relationship. A branch with moderate length was observed for Mascaruna, whereas the longest one was found for Orobica, Girgentana and Vallesana. The neighbor-net analysis using only the southern populations (Appendix A, 8POP) confirmed the results of the MDS described above and showed Maltese, Rossa Mediterranea and Nicastrese grouped into one cluster, whereas Girgentana, Aspromontana, Messinese and Argentata dell’Etna were grouped into another, with Mascaruna in an intermediate position.

The results of the admixture analysis agreed with the finding outlined above (Figure 6). We reported K values from 2 to 8 in order to underline ancestral components shared among different Italian goat populations. The model, assuming two ancestral populations (K = 2), separated the goats on a geographical axis, with the Italian northern populations (blue) assigned to a cluster and the Southern one assigned to a different cluster(red). Girgentana is one of the first breeds to separate out from Italian goat populations (K = 3, yellow), followed by Teramana (K = 4, green). For K = 5, the Mascaruna goat clustered apart from all other Italian goats, with some individuals showing a mixed ancestry. The lowest cross-validation error was recorded for K = 23. For K > 5 (up to 23 ancestral populations), the populations were progressively assigned to distinct clusters (Appendix A).

Finally, we used the TREEMIX software v.1.13 to model both population splits and gene flow using the whole dataset (32POP) and the genotypic data of Bezoar as outgroup (Figure 7). The *optM* function supported only one migration event (between Roccaverano and Maltese × Sarda); the graph shows a clear distribution of clusters according to the geographic origin of goat populations. However, the statistic calculated over five iterations for *m* from 0 to 10 indicated, at *m*9, a gene flow event between the base of the branch, including the Maltese populations and Mascaruna goat (Appendix A).

## 4. Discussion

The genetic diversity of autochthonous livestock populations is an important resource for food security and sustainable rural development. Moreover, indigenous populations are a valuable resource to preserve for their adaptation to harsh climatic conditions and resistance or tolerance to infection diseases common in their natural habitat [25]. However, the local animal genetic resources are poorly studied, and their management is neglected [26]. A genomic characterization of these populations represents an important tool to plan breeding programs and conservation strategies [27].

In this study, we performed the first genome-wide assessment of the genetic diversity and population structure of the local Mascaruna goat. The genetic diversity indices (such as Ho, He and F_IS_, Table 2) observed in this population are in concordance with the estimated values for Italian goats [16] and with most published studies employing Goat SNP50 BeadChip [6,9]. Moreover, the overall mean MAF is comparable to the results observed in other indigenous goat populations [28,29] and cosmopolitan breeds [11], suggesting that the impact of the ascertainment bias is small. The inferred CNe for Mascaruna was relatively low (34.4). This could be related to the geographic isolation of some farms. Overall, the results indicate moderate levels of genetic variability for Mascaruna goat.

Currently, the method based on ROH is considered one of the most powerful approaches to estimate genomic inbreeding [30]. Despite the absence or low frequency of male rotational schemes between farms, our results showed moderate level of genomic inbreeding for Mascaruna comparable to that of the other Italian local goat populations, such as Derivata di Siria, Facciuta and Nicastrese [16,31]. Estimates of ROH can be also used to give insights about population history [32]. The presence of a high percentage of short and medium ROH segments for Mascaruna is indicative of relatedness dating back to ancient times despite the absence of management for the mating plans. Moreover, only some individuals with more than 500 Mb of their autosomes covered by ROH were found in the Mascaruna. In order to minimize the loss in genetic diversity, those individuals could be excluded or less frequently used in mating plans. Monitoring and controlling inbreeding is important to limit the potential impact of deleterious alleles, inbreeding depression and loss of genetic variation. ROH in goat species were also used to identify genomic regions potentially under selection and involved in defining population-specific traits [16,30,33]. The ROH islands identified in the Mascaruna goat are linked to candidate genes involved in several biological functions. Within the ROH island on CHI2, we mapped *SATB2*, a gene associated with tick burden, with a potential link to the immune system [34]. On CHI6, we found a particularly interesting group of genes related to several economically important traits: *HERC5* and *HERC6* are linked with milk protein percentage in cattle [35] and *PPM1K* with total solids in milk [36]. Within this region, we mapped six important genes (*ABCG2*, *PKD2*, *SPP1*, *MEPE*, *IBSP* and *LAP3*) previously reported within a QTL related to milk production traits in cattle [37,38]. The ROH hotspot on CHI6 with the highest SNP occurrence (H = 45%) was mapped within known genes related to animal growth and development, such as *LCORL*, which was shown to regulate body size in goats [39], *FAM184B*, and *NCAPG* [40,41], associated with milk traits in Sicilian sheep [42]. Moreover, this genomic region overlapped with previously reported ROH islands in southern Italian goat [16,31], and Italian sheep breeds [43] and with selection signatures in local cattle [44]. Finally, in the ROH island on CHI12, we mapped *NBEA*, a gene involved in regulating body temperature in cattle with heat stress [45]. This gene, together with *DCLK1*, is reported as significantly associated with climatic variables in a landscape genomics study on Italian goat breeds [2] and, therefore, with a potential role for local adaptation. All these genes, related to milk production, immune response and local adaptation, are consistent with the phenotypic traits of the studied goat population.

Regarding population structure analyses, the results distinguished Mascaruna from other goat populations, indicating clear genetic differences compared with other Italian goats. The presence of a clear north–south geographical distribution of the genetic diversity (Figure 3) was highlighted by both the first two dimensions of the MDS plot and the neighbor-net, confirming the correspondence between geographical and genetic distances reported in previous studies on Italian goats [2,16]. These results were also supported by the low genetic differentiation (F_ST_) among populations from the same geographic area, which basically confirms this genetic structuring. Previously, a similar geographical pattern was described for Italian sheep [46] and cattle breeds [47]. The Admixture analysis corroborated these results and showed similar genetic background and shared ancestral components, grouping individuals according to the sampling locations. Such a partition was also supported by the TREEMIX graph, which indicated an agreement between clustering and geographic origin.

In general, the MDS separated Mascaruna from the other goat populations and, in particular, from northern and central Italian populations. Mascaruna goat formed a non-overlapping cluster and was clearly separated from the other populations. On the other hand, there were animals from officially recognized breeds that grouped together (Appendix A) and showed overlapping clusters, such as Argentata dell’Etna and Messinese or other goat populations from central Italy (Capra di Teramo, Fulva del Lazio, Capestrina, etc.). As expected, the Mascaruna clustered close to the southern Italian goats, towards which it showed the lowest genetic differentiation, especially to Argentata dell’Etna, which could be considered one of the ancestral populations. However, the Mascaruna diverged from Argentata dell’Etna and from the other Italian goat populations and is recognized as a distinct cluster at very low K value (K = 5), after historically differentiated breeds, such as Girgentana and Teramana. Concerning these two last breeds, our findings corroborated previous studies [1,2]. The results obtained on the genetic differentiation underline the hypothesis that Mascaruna goat could be a population with a defined genetic structure, especially if compared to other recognized breeds, despite the absence of a herd book and the possibility of influx from other populations. Such an interpretation agrees with information provided by breeders, who consider the Mascaruna as an independent goat population, reared separately and without crossbreeding with the other Sicilian goats. In confirmation of this, only one migration event was detected between Mascaruna and the Maltese breed, and it was not identified in the most supported event (*m1*). Therefore, it is likely that this population experienced reproductive isolation and reduced gene flow and, thus, acquired a genetic identity.

To explore in detail the relationship with the other goat populations of southern Italy, a further analysis was carried out considering only Mascaruna and these populations. The results confirmed the differentiation between Mascaruna and southern populations, which could be hypothesized as its potential ancestors, as above reported. Compared with these populations, the Mascaruna goats are relatively more dispersed within their cluster in MDS, which is typical of population that show higher genetic diversity or admixture with other breeds [48]. In general, the Northern populations showed clearly recognizable clusters as a consequence of reproductive isolation and reduced gene flow, whereas the Southern goat populations showed admixture patterns [2]. Within the Mascaruna cluster, some individuals showed patterns of admixture, similar to other individuals belonging to officially recognized breeds (e.g., Grigia Ciociara, Garganica) (Appendix A). Conversely, the moderate levels of admixture in several Mascaruna goats indicated that there were fewer genetic components remaining from any other ancestral breed that may have interacted with it, even if this may be also considered a typical signal of inbreeding [46]. However, the population did not show high levels of inbreeding, as indicated by the moderate values of genetic diversity and the branch length in the neighbor-net. However, the ancestral origin of the Mascaruna remains uncertain. Humans often have introduced livestock into islands to be used as a source of food by settlers. The magnitude, timing and number of such founder effects may have had long-term consequences on the diversity of insular populations [49]. In the absence of migration, founder events featured by a few individuals will produce populations that show, irrespective of their current size, a decreased variability and a marked genetic differentiation due to the impact of drift [49]. Therefore, the differentiation of Mascaruna goat is probably the consequence of the combined effects of genetic drift, small population size, founder effects and reproductive isolation. Further study will be necessary to obtain a fine-grained perspective of the ancestral component of this goat population.

Recently, Dadousis et al. [50], taking into consideration the concept of breed, reported an interesting definition from Hammond [51]: “*A breed is a breed if enough people say it is*”. Historical information provided by farmers indicated that the Mascaruna was reared in Sicily since the Second World War, and is widely known among local goat farmers, who consider it an official breed. Several studies reported a weak structuring of goat populations and showed the perspective that many so called breeds are actually landraces at best and panmixia predominates in these genetic reservoirs [52,53,54]. Lenstra et al. [55] suggested that pure genetic ancestry was not a prerequisite for goat breeds. Local livestock populations, generally consisting of small populations reared in specific regions, often face difficulties in obtaining formal recognition as a breed [50]. Therefore, based on these considerations and the multiple pieces of evidence outlined in this study, our findings represent a starting point for the creation of monitoring and conservation plans aimed at defining a possible official recognition program as a breed. However, an additional analysis and a wider sampling would contribute to refine and validate these results.

## 5. Conclusions

In this study, we reported the first genome-wide results on the genetic diversity and population structure of the local Mascaruna goat. The results indicate the existence of a defined genetic background to safeguard. For conservation in farms, breeders need to be financially supported by local and international authorities [49], and this will be possible after an official recognition of the Mascaruna goat as a breed. Subsequently, a direct link between the Mascaruna and the production of niche products could add an extra economic value [10]. Finally, our study describing the diversity and structure of Mascaruna confirms the usefulness of applied genomic analyses as valid tools for the study of the local uncharacterized genetic resources.

## Figures and Tables

**Figure 1 animals-12-02840-f001:**
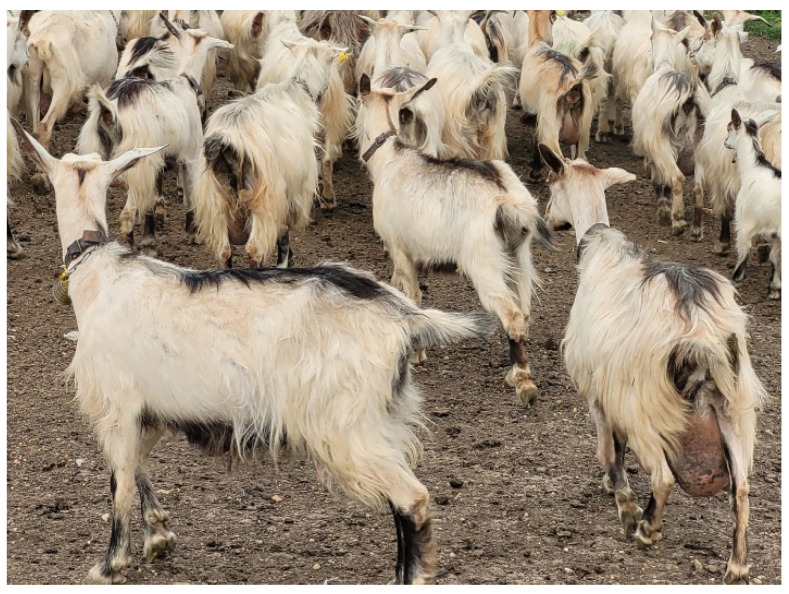
Mascaruna goats.

**Figure 2 animals-12-02840-f002:**
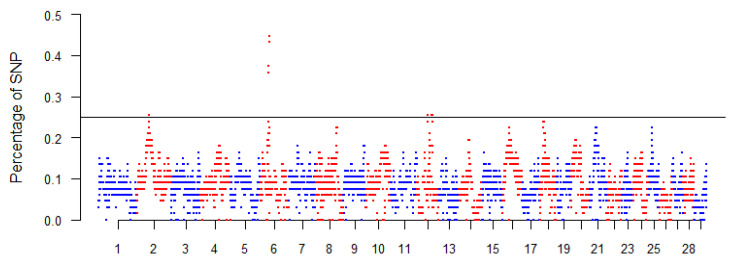
Manhattan plot of the incidence of each SNP in the runs of homozygosity in Mascaruna goat. A threshold of 0.25 was chosen to detect the islands.

**Figure 3 animals-12-02840-f003:**
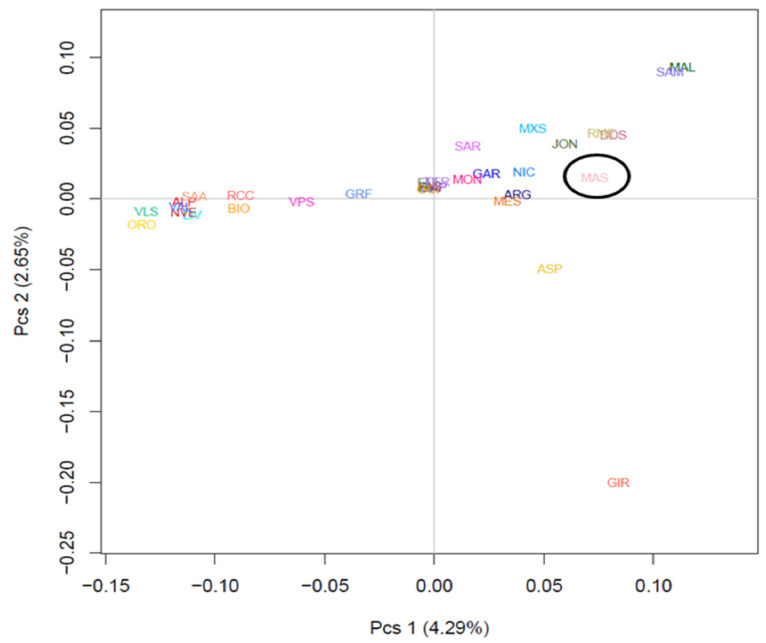
Multidimensional scaling analysis among 32 goat populations using the average coordinates of eigenvalues of C1 and C2. For the full definition of populations, see Table 1 (Mascaruna = MAS).

**Figure 4 animals-12-02840-f004:**
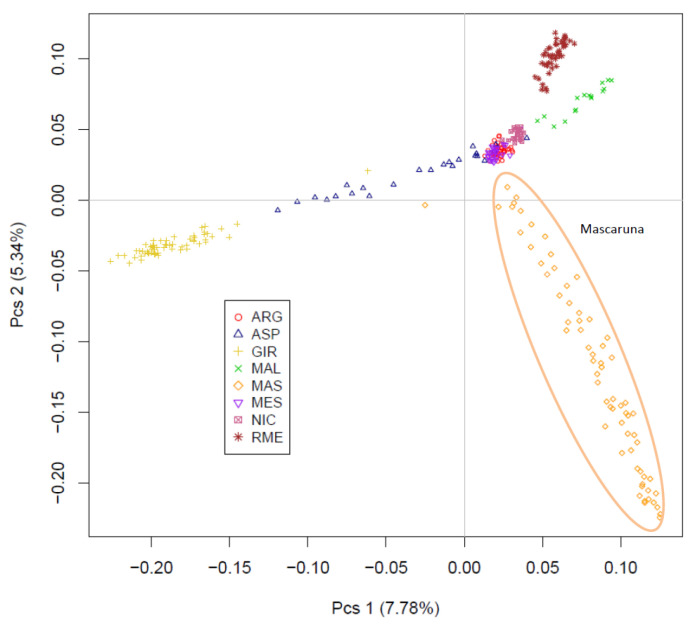
Multidimensional scaling analysis between Mascaruna and the southern goat populations. For full definition of the populations, see Table 1.

**Figure 5 animals-12-02840-f005:**
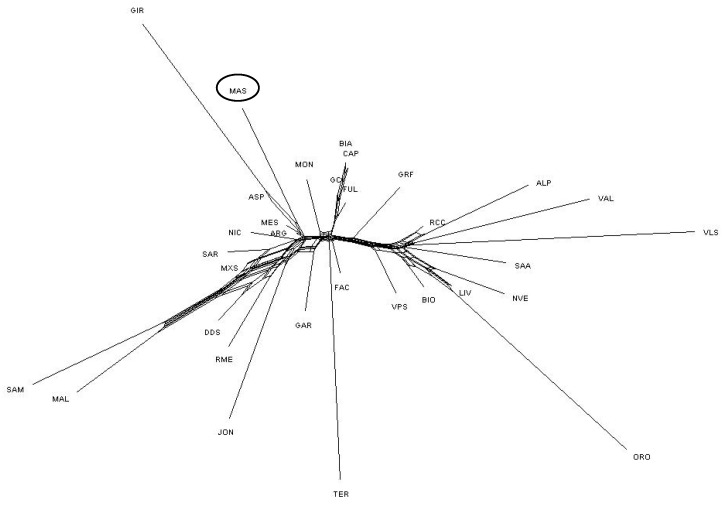
Neighbor-net graph based on Reynolds genetic distances among 32 goat populations. For the full definition of populations, see Table 1 (Mascaruna = MAS).

**Figure 6 animals-12-02840-f006:**
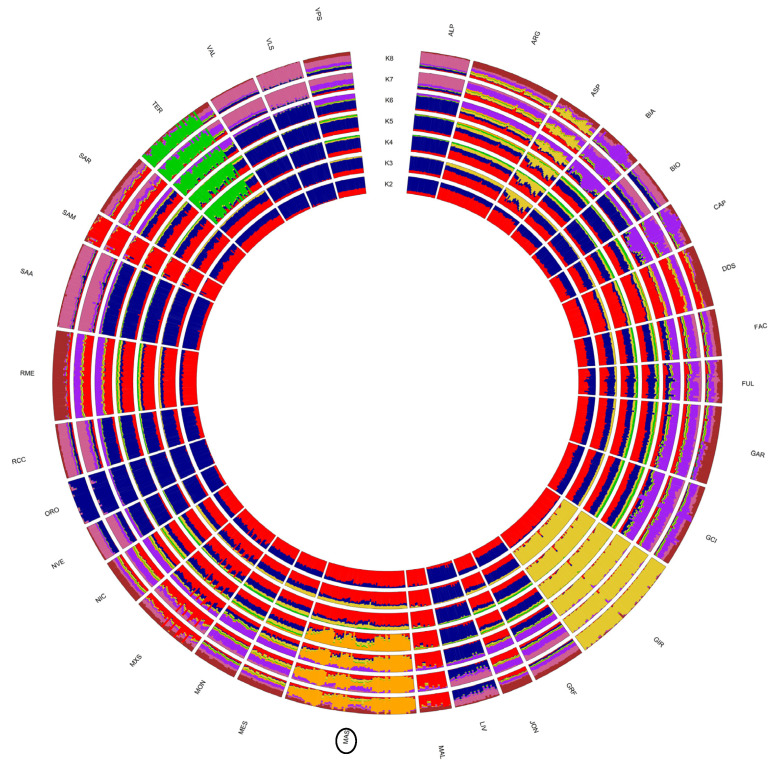
Population structure for 32 goat populations inferred from the ADMIXTURE analysis with K values from 2 through 8. For the full definition of populations, see Table 1 (Mascaruna = MAS).

**Figure 7 animals-12-02840-f007:**
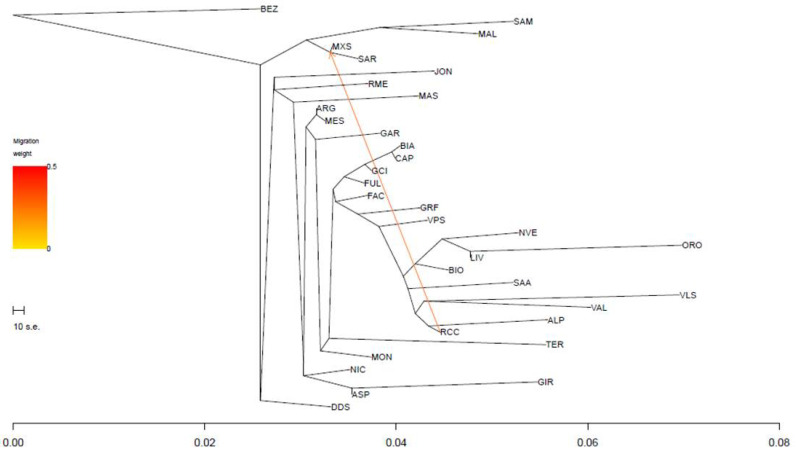
TREEMIX analysis with the most supported number of migration events (*m* = 1). For the full definition of populations, see Table 1 (Mascaruna = MAS).

**Table 1 animals-12-02840-t001:** Acronym, population name, number of individuals after quality control (N) used in the analyses.

Acronym	Population Name	N	32POP	8POP
ALP	Camosciata delle Alpi	25	X	
ARG	Argentata dell’Etna	46	X	X
ASP	Capra dell’Aspromonte	24	X	X
BEZ	Bezoar	7		
BIA	Bianca Monticellana	24	X	
BIO	Bionda dell’Adamello	24	X	
CAP	Capestrina	22	X	
DDS	Derivata di Siria	32	X	
FAC	Facciuta della Valnerina	24	X	
FUL	Fulva del Lazio	22	X	
GAR	Garganica	40	X	
GCI	Grigia Ciociara	43	X	
GIR	Girgentana	59	X	X
GRF	Garfagnina	27	X	
JON	Jonica	16	X	
LIV	Capra di Livo	23	X	
MAL	Maltese	16	X	X
MES	Messinese	24	X	X
MAS	Mascaruna	67	X	X
MON	Capra di Montefalcone	23	X	
MXS	Maltese e Sarda	36	X	
NIC	Nicastrese	24	X	X
NVE	Nera di Verzasca	19	X	
ORO	Orobica	23	X	
RCC	Roccaverano	28	X	
RME	Rossa Mediterranea	46	X	X
SAA	Saanen	44	X	
SAM	Maltese sampled in Sardinia	15	X	
SAR	Sarda	33	X	
TER	Capra di Teramo	43	X	
VAL	Valdostana	24	X	
VLS	Vallesana	23	X	
VPS	Capra della Val Passiria	24	X	

**Table 2 animals-12-02840-t002:** Estimates of genetic diversity indices in Mascaruna goat.

Ho ± s.d.	He ± s.d.	MAF ± s.d.	F_IS_ ± s.d.	F_ROH_± s.d.	CNe
0.398 ± 0.122	0.401 ± 0.104	0.313 ± 0.121	0.008 ± 0.087	0.084 ± 0.080	34.4

Ho = observed heterozygosity; He = expected heterozygosity; F_IS_ = inbreeding coefficient based on the difference between observed vs. expected number of homozygous genotypes; MAF = average minor allele frequency; F_ROH_ = inbreeding coefficient based on runs of homozygosity; s.d. = standard deviation; CNe = contemporaney effective population size.

**Table 3 animals-12-02840-t003:** Runs of homozygosity islands identified in Mascaruna goat.

CHI	Start (bp)	End (bp)	n SNPs	Genes
2	47,507,659	48,088,038	12	*SATB2*
6	36,764,986	38,590,195	31	*HERC5*, *LOC106502196*, *HERC6*, *LOC108636216*, *PPM1K*, *ABCG2*, *PKD2*, *SPP1*, *MEPE*, *IBSP*, *LOC102176111*, *TRNAA-CGC*, *LAP3*, *MED28*, *FAM184B*, *LOC106502187*, *DCAF16*, *NCAPG*, *LCORL*, *TRNAC-GCA*
12	43,807,522	44,611,581	15	*LOC102173756*
12	60,001,083	61,020,592	17	*LOC102178917*, *TRNAC-GCA*, *NBEA*, *LOC102175490*, *MAB21L1*, *TRNAE-UUC*, *LOC102178636*, *DCLK1*

CHI = *Capra hircus* chromosome. n SNPs = number of single nucleotide polymorphisms.

## Data Availability

The data that support the findings of this study are available on request from the corresponding author.

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
