# Peer review of "Genomic Tools for the Characterization of Local Animal Genetic Resources: Application in Mascaruna Goat"

_animals, 2022, doi:10.3390/ani12202840_

Round 1
Reviewer 1 Report (Previous Reviewer 1)
The authors have addressed my comments. Some minor suggestions are below
Line 88-91: Which tissues did the authors use for DNA isolation, and which protocol was used?
How many farms were used in the current study?
Table 3: N SNPs, please write full name "Number of SNPs"
Author Response
REVIEWER 1
The authors have addressed my comments. Some minor suggestions are below
Line 88-91: Which tissues did the authors use for DNA isolation, and which protocol was used?
How many farms were used in the current study?
Table 3: N SNPs, please write full name "Number of SNPs"
RESPONSE: thanks for the general comment and suggestions. The required information has been added in the revised manuscript (tissue used, protocol, number of farms).
In the Table 3 we as added a full explanation of N SNPs, below the Table.
Reviewer 2 Report (Previous Reviewer 3)
The paper gives information on the genetic variability of a population of goats raised in the Sicily region (Mascurana local goat population). This information may be useful for the maintenance of endangered genetic groups/breeds
Author Response
The paper gives information on the genetic variability of a population of goats raised in the Sicily region (Mascurana local goat population). This information may be useful for the maintenance of endangered genetic groups/breeds
RESPONSE: thanks for the general positive comment.
Reviewer 3 Report (Previous Reviewer 2)
In this study, the authors genotyped 72 Mascaruna goats using Illumina Goat_IGGC_65K_v2BeadChip and analyzed genome-wide data with other Italian goat populations with the aim of studying the epigenetic diversity of Mascaruna goats. This study has a clear intention, clear thinking and a rich article. But there are still some things to think about.
1 Simple Summary: and Abstract and in the full text, "a defined population structure and genetic potential to start a process of official recognition as breed ," and "and our findings represent a starting point for the creation of a conservation plans" are a bit exaggerated. In this paper, we have only drawn some conclusions from the data analysis, and we need to expand the population to verify the conditions for recognition as breed after extensive analysis.
2 Since this is a joint analysis with data from Italian goats, shouldn't there be a clear contrast shown in the material methods and in the results and discussion?
3 Is it possible to conclude genetic variability using 72 Mascaruna goats? The population should be expanded to calculate genetic variability.
4 "3.2Runs of homozygosity islands in Mascaruna goat" should develop an exhaustive description of Figure 2.
5 The discussion section should be integrated into a coherent paragraph. Writing a separate topic for each subsection does not show the rigor and logic of the essay.
Author Response
In this study, the authors genotyped 72 Mascaruna goats using Illumina Goat_IGGC_65K_v2BeadChip and analyzed genome-wide data with other Italian goat populations with the aim of studying the epigenetic diversity of Mascaruna goats. This study has a clear intention, clear thinking and a rich article. But there are still some things to think about.
RESPONSE: thanks for the general comment and suggestions.
1 Simple Summary: and Abstract and in the full text, "a defined population structure and genetic potential to start a process of official recognition as breed ," and "and our findings represent a starting point for the creation of a conservation plans" are a bit exaggerated. In this paper, we have only drawn some conclusions from the data analysis, and we need to expand the population to verify the conditions for recognition as breed after extensive analysis.
RESPONSE: thanks; following the reviewer’s suggestions, we modified the sentences in simple summary, abstract and in full text.
For example, in the revised abstract, we have reported this sentence: “Although they are not conclusive, our current results represent a starting point for the creation of monitoring and conservation plans. Additional analyses and a wider sampling would contribute to refine and validate these results”.
2 Since this is a joint analysis with data from Italian goats, shouldn't there be a clear contrast shown in the material methods and in the results and discussion?
RESPONSE: the information for the genotypic data of Italian Goat Consortium2 (IGC2) used in this study is reported in material and methods section, and a full list of goat populations is reported in Table 1. We used the dataset to study the genetic relationships, patterns of ancestry and admixture of Mascaruna in the Italian context. The finding of the analyses are reported in the results and discussion sections.
3 Is it possible to conclude genetic variability using 72 Mascaruna goats? The population should be expanded to calculate genetic variability.
RESPONSE: thanks; following the reviewer’s comment in the revised manuscript we added that additional analyses and an increase in the number of genotyped animals would be particularly relevant to refine and validate these results.
However, we think that 72 individuals is a representative sample (or at least sufficient) to estimate the genetic diversity indices for a population and to provide a first estimate. As reported in the manuscript, we collected animals as much unrelated as possible, to evaluate the genetic diversity and avoid to analyze related animals. It should also be emphasized that it is a local genetic resource with a reduced population size.
4 "3.2 Runs of homozygosity islands in Mascaruna goat" should develop an exhaustive description of Figure 2.
RESPONSE: Thanks for the suggestion. We added a further description of Figure 2. Table 3 also help to understand the position of the islands in the Manhattan plot.
5 The discussion section should be integrated into a coherent paragraph. Writing a separate topic for each subsection does not show the rigor and logic of the essay.
RESPONSE: thanks for suggestion. The discussion section is reported as single paragraph in the revised manuscript.
Round 2
Reviewer 3 Report (Previous Reviewer 2)
None.
This manuscript is a resubmission of an earlier submission. The following is a list of the peer review reports and author responses from that submission.
Round 1
Reviewer 1 Report
Overall, the manuscript is well written and provides exciting results for conserving the Mascaruna goat. The approaches are reasonable as the authors can use other data from ICG2 for a better understanding of the diversity and population structure of the breed. I believe that the manuscript is useful for the scientific community working on conservation biology as well as practical aspects of the conservation of the breed.
Line 118: The authors might provide parameter settings for ROH identification here.
Line 128-129: The authors might give the references or links for the tool.
Line 162: Table 2 might not be necessary as the authors can write down the information in a text.
Line 175: Why did the authors choose the threshold of 0.999%
Line 87 and 104-107: What are the sexes of these animals and did the authors remove the sex chromosomes in the QC for SNPs?
Reviewer 2 Report
1 The sample was from a total of 72 Mascaruna goats from 10 farms, which is too small to adequately describe genetic polymorphism, population structure And the effect of different farms on the goats was not taken into account.
2 It cannot be sufficiently described that Mascaruna goats can be a new breed, and a lot of work is needed to verify this conclusion.
Reviewer 3 Report
The manuscript contains information about the structure of the Mascaruna goat population.
The methods are adequately described and the exposition is clear.
I have only one suggestion for the authors,
for this type of study, the sampling of the animals must be done in order to reduce the genetic relationship between the animals of various farms and thus increase the representativeness of the group of animals studied.
In the manuscript the geographical areas of the Sicily region with the farms where the animals were sampled should be reported (for example adding a map of the Sicily region or entering the geographical coordinates of the farms)